# Parents’ Perception, Acceptance, and Hesitancy to Vaccinate Their Children against COVID-19: Results from a National Study in the UAE

**DOI:** 10.3390/vaccines10091434

**Published:** 2022-08-31

**Authors:** Zelal Kharaba, Rahaf Ahmed, Alaa M. Khalil, Raneem M. Al-Ahmed, Amira S. A. Said, Asim Ahmed Elnour, Sarah Cherri, Feras Jirjees, Hala Afifi, Naglaa S. Ashmawy, Bassam Mahboub, Yassen Alfoteih

**Affiliations:** 1Program of Clinical Pharmacy, College of Pharmacy, Al Ain University, Abu Dhabi Campus, Abu Dhabi P.O. Box 112612, United Arab Emirates; 2AAU Health and Biomedical Center, Al Ain University, Abu Dhabi P.O. Box 112612, United Arab Emirates; 3Faculty of Medical Sciences, Newcastle University, Newcastle upon Tyne NE2 4HH, UK; 4Clinical Pharmacy Department, College of Pharmacy, Al Ain University, Al Ain P.O. Box 64141, United Arab Emirates; 5Clinical Pharmacy Department, Faculty of Pharmacy, Beni Suef University, Beni Suef P.O. Box 64141, Egypt; 6School of Pharmacy, Lebanese International University, Mouseitbah, Beirut P.O. Box 146404, Lebanon; 7College of Pharmacy, University of Sharjah, Sharjah P.O. Box 27272, United Arab Emirates; 8Department of Pharmacy, City University College of Ajman, Ajman P.O. Box 18484, United Arab Emirates; 9Department of Pharmaceutical Chemistry, Faculty of Pharmacy, Ain-Shams University, Cairo P.O. Box 11566, Egypt; 10Department of Pharmacognosy, Faculty of Pharmacy, Ain Shams University, Cairo P.O. Box 11566, Egypt; 11Rochester Institute of Technology-Dubai, Dubai P.O. Box 341055, United Arab Emirates; 12Department of Respiratory Medicine, Rashid Hospital, Dubai Health Authority, Dubai P.O. Box 4545, United Arab Emirates; 13Sharjah Institute for Medical Research, College of Medicine, University of Sharjah, Sharjah P.O. Box 27272, United Arab Emirates; 14Department of Dental Surgery, City University College of Ajman, Ajman P.O. Box 18484, United Arab Emirates; 15Department of General Education, City University College of Ajman, Ajman P.O. Box 18484, United Arab Emirates

**Keywords:** COVID-19, children vaccination, parents’ perception, distance learning, vaccine hesitancy, national study

## Abstract

**Introduction:** COVID-19 is considered the greatest health disaster affecting humans during the 21st century, which urged the need to develop an effective vaccine to acquire enough immunity against the virus. The main challenge faced during the development of such vaccines was the insufficiency of time, which raised the question about the vaccine safety and efficacy, especially among children. Parents’ and caregivers’ thoughts and acceptance of administering the vaccine to their children are still debatable topics and are yet to be explored in the UAE. **Aims:** The study aims to exploit parent acceptance, perception, and hesitancy toward the COVID-19 vaccine administration for their children and the link with their choice of distance learning instead of face-to-face education in the UAE. **Methodology:** This study utilized a cross-sectional descriptive design. A sample of 1049 parents across all emirates were conveniently approached and surveyed using Google forms from June to September 2021. The participants responded to a semi-structured questionnaire pertaining to socio-demographic, educational, and other questions related to COVID-19 and its link with their beliefs in whether the vaccination of their children will help with resuming face-to-face learning. **Results**: Approximately 74% of the parents confirmed that their children who are 16 years old and above have received the vaccine, and 71% were willing to give the vaccine to their children aged above 5 years. Parents with children receiving online education and those with children where the online modality of learning negatively affected their academic achievement are more prone to administer the COVID-19 vaccine to their children above five years old. The results show a significant association between vaccination of children and the parental desire for resuming physical attendance in schools (*p* value < 0.001). Multivariate analysis showed that the highest acceptance rate was from parents with children of low academic achievement due to online learning modality during the pandemic. **Conclusion:** In the UAE, parents of young children have shown a positive attitude towards COVID-19 vaccination in belief that vaccines will reduce the risk of infection and assist in resuming normal lifestyles, such as going back physically to schools. The results reflect the public awareness and the hypervigilance regarding the COVID-19 pandemic in the UAE.

## 1. Introduction

The new COVID-19 outbreak took the whole world by surprise since the reporting of the first case in Wuhan, China, on the 31 December 2019. The highly contagious virus and the associated morbidity and mortality rates have alarmed the whole world and since then health authorities and countries have tried to adopt different preventive measures to contain the virus, which has infected around 582 million people worldwide and resulted in approximately 6.4 million mortalities as of August 2022 [1]. This new virus has resulted in a pandemic and affected many aspects of society, such as the economic market, lifestyles, and the healthcare sector. Preventive measures that aimed to reduce the infection rates were adopted worldwide, such as keeping a 3 feet social distance between individuals, mandating face masks, quarantining for infected individuals and close contacts, and closing facilities that were not mandatory to sustain life. Although these measures helped in reducing the viral exposure, they were insufficient to repress the virus, as the infected cases continued to increase. Therefore, the necessity to develop herd immunity was evident to overcome this pandemic. Many studies documented the difficulties of achieving herd immunity through natural infection [2]. Therefore, establishing herd immunity through vaccination programs to prevent further transmission of the virus was prioritized by many countries [3]. In addition, healthcare professionals and authorities agreed on the importance of vaccination to combat the COVID-19 pandemic, as it reduces the risk of viral infection and the severity of the experienced symptoms by patients. Although vaccination programs represented a very optimistic approach to end the COVID-19 pandemic, the hesitancy, criticism, and the fear of vaccinations among the population worldwide, especially for children, were the major obstacles facing these programs [4]. The public perceptions and willingness to be vaccinated are key determinants for the success of the vaccination program and, hence, the control of the pandemic.

The UAE was among the first countries to adopt measures to prevent the spread of the virus. Soon after declaring COVID-19 as a pandemic, the UAE imposed a national quarantine setting for the whole population, regulations on wearing face masks and fines for removing them, as well as limiting all social gatherings. The UAE imposed strict regulations on the entry to the country and the use of indoor facilities, which became only allowed for green pass holders on the “Al-HOSN” application. The application is a national official app for contact tracing, vaccination status, and health information related to COVID-19 [5]. As an attempt to provide excellent healthcare service and protect the UAE citizens and residences against the threatening pandemic, the UAE provided all students, members of the healthcare workforce, and public service providers free PCR screening, in addition to a free national vaccination program for all UAE citizens and residents. Currently, five vaccines are available in the UAE, Sinopharm, Pfizer-BioNTech, Sputnik V, Oxford-AstraZeneca, and Moderna [5].

COVID-19 is a virulent virus which infects children and elderly members of the population equally, especially with the recently identified highly contagious variants (Omicron and Delta). Adult vaccination programs have shown remarkable success; however, child vaccination has been lagging, which contributed to the continuity of the high infection rates [6]. Thus, even with children not showing severe symptoms of infection, they act as vectors that contribute to the transmission of the virus to other children and adults [7]. Moreover, studies have proven that children might experience more severe symptoms and long-term health impacts such as excessive weariness, shortness of breath, memory loss, and academic underperformance after recovering from the illness [8,9].

According to recent emerging information regarding the COVID-19 vaccine and its suitability for children, the best course of action was to start the implementation of the child vaccination programs. The UAE has not spared any efforts to protect its citizens and residents and became among the first countries worldwide to adopt child vaccination programs after the FDA approval. Currently, Sinopharm and Pfizer-BioNTech vaccines are available for children of different age groups in the UAE [10]. The UAE started launching initiatives urging parents to get children vaccinated against COVID-19 [11], with an expected high level of compliance.

The merging child vaccination programs have experienced a high amount of speculation and concern from parents with conflicting information and beliefs regarding the vaccine safety and efficacy for individuals of young age. The scarcity of studies that discussed the rate of child vaccination against COVID-19 worldwide and the associated safety issues have helped in the rise of resistance against vaccinating children. One Italian cross-sectional study evaluated the knowledge, attitude, and intention to vaccinate children <18 years. The study concluded that around 40% of families of children aged ≥12 years and around 36% among parents of children <12 years would not vaccinate their children [12]. In addition, there is a lack of studies regarding COVID-19 vaccine hesitancy issues. One systematic review discussed the hesitancy towards COVID-19 vaccine among the public [13]. In another study, vaccine hesitancy among four African and Middle Eastern countries was found to be approximately 30% [14].

To the best of our knowledge, this study is the first nationwide research in the UAE with a representative sample size to address the parents’ perception regarding child vaccination. The aims of this study were to explore parents’ acceptance, perception, and hesitancy toward COVID-19 vaccine administration for their children, and the association with their choice of distance versus face-to-face learning modality for their children.

The study provides important data to the scientific community and the healthcare sector in the UAE and the world related to reasons of reluctance among parents to give the COVID-19 vaccine to their children. This study will contribute to determining the impact of vaccination on the choice of the educational modality. Furthermore, the results will help the health authorities to prepare a plan to educate people about the vaccine for children, specifically if the vaccine will be more in demand for young children (less than 5 years) in the future.

## 2. Methodology

### 2.1. Study Design, Sampling Technique, and Size

A descriptive questionnaire-based cross-sectional online survey was employed in this study across all emirates in the UAE (*n* = 1049). Participants were parents aged above 18 years, who gave their consent to participate in the study. The research ethics commission (REC) of Al Ain University granted the study the necessary ethical permissions (AAU-REC-B3, February 2021). Convenience sampling was employed to recruit participants for the study. Based on previous literature and on the Raosoft sample size calculator [15], a minimum sample size of 385 students was considered as a representative sample for this study. Participants were approached through social media platforms. The questionnaire was presented in both English and Arabic languages to permit the participation of Arab and non-Arab parents in the UAE. The questionnaire was developed in English language and then translated into Arabic language by two bilingual specialists. The questionnaire was uploaded to Google Form^®^ and distributed to those who could access the online survey. The inclusion criteria of participants were living in one of the seven emirates of the UAE for more than 2 years, aged 18 or older, married with children, understand Arabic or English language, and agree to fill the online questionnaire. The responses of single participants or those married with no children were excluded from this study. The online link of the survey, including a consent form, was sent to participants via media platforms such as WhatsApp^®^, Instagram^®^, Facebook^®^, Twitter^®^, and email, for a period of four months: June to September 2021. Respondents were asked to share the questionnaire with their friends, relatives, and social networks. Furthermore, the respondents were notified about the policy of confidentiality and anonymity.

### 2.2. Study Tool

The available literature was intensively reviewed, and a questionnaire was developed (S1) to be used in this study. The semi-structured questionnaire is an approximately 5 min questionnaire which includes socio-demographic, lifestyle, and health awareness questions related to the COVID-19 pandemic, such as the participants acceptance for the COVID-19 vaccination and whether they will administer it to their children, the link between the mode of education and child vaccination, in addition to the source of information regarding the COVID-19 pandemic and the vaccine consulted by the parents.

### 2.3. Characteristics of Participants

The following information was addressed: age, gender, residence area, education level, study field, and employment status.

### 2.4. Parents’ Perspective towards the COVID-19 Vaccine

Participants answered several Yes/No questions about their knowledge, acceptance, and concerns towards the COVID-19 vaccine in general and towards children vaccination in particular. Five Likert scale questions were used to address the parents’ decision to send their children back to school and their concern regarding the vaccine side effects. Most of the questions were designed to seek the parents’ preference for future learning modality for their children. The options given were either to select the distance or face-to-face learning mode to establish a relation between the children’s COVID-19 vaccine and returning to face-to-face learning mode.

### 2.5. Validation of the Study Questionnaire

The questionnaire used in the study is semi-structured. A draft of the questionnaire was prepared and sent to a panel of experts and professors in the clinical pharmacy departments at Al Ain University, Ajman University, and UAE University to test the validity of the questionnaire’s content. The validity of the questionnaire’s content comprises many factors such as the conciseness, length, clarity, language, time consumed, bias, and appropriateness of questions [16].

To develop and validate the Arabic version of the questionnaire, a forward translation procedure was performed. First, the English survey questionnaire was translated into Arabic by an independent bilingual (Arabic/English) translator. The version was subsequently retranslated into its original language by another independent bilingual translator who was not familiar with the original. Then, the translated version was reviewed by three authors.

### 2.6. Reliability Testing of the Study Questionnaire

The questionnaire was revised based on a reliability test conducted as a pilot study on 30 participants to achieve the most acceptable Cronbach’s values (0.76). In addition, preliminary pilot testing was performed to ensure the practicality and understandability of the questionnaire.

### 2.7. Statistical Analysis

Data sets were gathered and entered on an Excel sheet, which was imported to the Statistical Package for Social Science (SPSS) version 26. Descriptive statistics were used to describe patient characteristics with frequencies and percentages for categorical variables and mean ± standard deviation for continuous variables. Univariate associations were assessed for statistically significant differences using Chi-square or Fisher’s exact tests for categorical variables, as appropriate. Logistic regression was performed to identify factors associated with the parents’ acceptance of the COVID-19 vaccine and the parents’ willingness to get their children back to school. Multivariate analysis was performed to evaluate the relationship between various independent variables and the parental acceptance of the COVID-19 vaccine for children as a dependent variable and parental willingness to send their children back to school as a dependent variable. The included independent variables are the variables used in the bivariate analysis. The results were presented as odds ratios (OR) and 95% CI. All reported *p* values were two-sided with the alpha set at a significance of 0.05. The statistical analysis adjusted for the missing values by accounting for the valid percentage in the results output.

## 3. Results

### 3.1. Socio-Demographic Characteristics

A total number of 1049 participants were enrolled in this study, most of them were in the age category 18 to 30 years (32.0%), with 72.1% being females. More than half the participants (50.6%) hold a bachelor’s degree and about half of them (49.0%) work in non-healthcare sectors (Table 1).

### 3.2. Parents’ Perception towards COVID-19 Vaccine

Approximately 74% of the parents confirmed that their children above the age of 16 have received the vaccine and 71% are willing to administer the vaccine to their children aged above five years, upon its availability (Table 2).

In terms of educational mode of children, as per the parents, 71% of the children are attending their classes online. Two-thirds of the parents believe that online mode has negatively affected their children academic performance. When parents were asked about physical attendance of children in schools, approximately 65% of the parents believed that it is associated with health risks (risk > 3). However, 71% of the parents agree that vaccinated children will be safe physically return to schools (Table 2).

The results demonstrated in Table 2, concluded that 71% of the participants are still concerned about the pandemic. 60% of the parents confirmed that their children are vaccinated against the seasonal flu and follow the safety protocols in regard to wearing masks in public areas (83%) and self-isolation of children showing any relevant symptoms (79%). Furthermore, 50% of the participants shared that social media is their main source of information regarding the COVID-19 virus and the recommended protective measures. 

### 3.3. Association between Socio-Demographic Characteristics and Parental Perception towards COVID-19 Vaccine with Parental Acceptance of the COVID-19 Vaccine for Children

Male parents were the majority among the accepting participants (84%), regardless of their age group, and there was 60% acceptance of the vaccine across all age groups. The educational level showed significant difference in result with comparable values between the groups (*p* = 0.003). Healthcare workers reported the highest percentage of acceptance of approximately 87% (*p* < 0.001) (Table 3). 

### 3.4. Association between Parents’ Perception towards COVID-19 Vaccine and Parents’ Acceptance of the COVID-19 Vaccine

A significant association is reported between the percentage of acceptance of children vaccination and the presence of elderly individuals within the household (*p* < 0.001). The perception that COVID-19 still imposes risks to our health and that vaccination could protect the children and enable them to return physically to school created a strong relationship (*p* < 0.001). Friends and social networks were the main source of information regarding the vaccine efficiency among those who accepted its administration their children above 5 years old (*p* value < 0.001) (Table 4).

### 3.5. Association between Socio-Demographic Characteristics and Parental Perception towards COVID-19 Vaccine with Parental Willingness to Send Their Children Back to School

The association between socio-demographic characteristics with parenteral willingness to send their children back to school reported that male participants within the age range of 31 to 40 have a significantly higher tendency to give the COVID-19 vaccine to their children above the age of 5 (*p* value < 0.001). In addition, participants who work in the medical field demonstrate a remarkable association, with a *p* value < 0.001. The results are shown in Table 5.

Table 6 demonstrates the association between different variables and the parental willingness to send their children back to school instead of online classes. A significant association between physical attendance in schools and parents’ agreement to vaccinate their children (>5 years) against COVID-19, was reported. A similar pattern was observed among participants with children whose academic achievement has been negatively impacted (*p* value < 0.001). Furthermore, most of the participants strongly supported the return of children to in-person education (*p* value < 0.001).

### 3.6. Multivariate Analysis

Results of the multivariate analysis (Table 7) concluded that the highest percentage of acceptance of COVID-19 vaccination for children was reported by the female participants (OR = 0.403; *p* value = 0.002). Healthcare workers have shown higher acceptance rate compared to retired participants, students, and unemployed individuals (*p* value = 0.039, 0.009, and 0.002, respectively). In addition, a remarkable high acceptance rate was observed among participants with vaccinated family members aged below or above 16 years. These participants believe that vaccination will protect their children from infection, in addition to other preventive measures such as the flu vaccine and wearing face masks in public areas (*p* value = 0.029, 0.002, <0.001, 0.044, and 0.001, respectively).

Results of the multivariate analysis (Table 8) reported the highest tendency for children vaccination among parents aged between 18 and 30 years, when compared to those aged 41 and above, as they are eager to send their children back to school (*p* value = 0.024, and 0.002, respectively). Notwithstanding, participants with a higher level of education recorded a remarkably higher willingness to resume physical education for their children compared to the undergraduate participants (*p* value = 0.025, and 0.003, respectively). Furthermore, remarkable high values were shown with parents of children with low academic achievement due to online learning during the pandemic and those who believe that vaccination of children would aid in their physical returning to schools. Similar trend was reported among parents who gave their children the flu vaccine and urge them to wear face masks in public areas (*p* value = 0.006, <0.001, <0.001, 0.018, and 0.081, respectively) (Table 8).

## 4. Discussion

The study aimed to investigate parents’ perception and acceptance of the available COVID-19 vaccine, their willingness to administer it to their children and the associated factors. The total of 1049 participants were enrolled in this study, most of whom were female, and more than half of them had a bachelor’s degree. Three-quarters of participants confirmed that their children over the age of 16 had received the vaccine, and more than two-thirds of them were willing to give the vaccine to their children over the age of 5. Parents with children receiving online education and are willing to send their children back to school have shown a high acceptance rate for children (>5 years) vaccination against COVID-19. The highest acceptance rate was observed among parents with children who had low academic achievement due to the online learning method during the pandemic.

Most of the young participants (18–32 years) support the idea of children vaccination against the COVID-19 virus, which can be explained by the advanced education level of the participants and their accessibility to trustworthy sources of information. This finding came in agreement with a previous study on the Jordanian population, which reported higher vaccine acceptance among younger participants [3]. On the other hand, in USA and Saudi Arabia, studies demonstrated a higher acceptance rate among the older generations, which could be due to demographic and geographical discrepancies [17,18,19].

The reported results have concluded that females are more supportive of COVID-19 vaccination programs compared to the male participants, which supports the general direction of female perception worldwide, as shown in other GCC countries such as Saudi Arabia and globally in another 19 countries [18,20].

Surprisingly, the acceptance rate for children vaccination against COVID-19 among healthcare workers in the UAE was remarkably high compared to previously published data (>70% of the participants, *p* < 0.01). Previous studies reported notable reluctancy of healthcare workers towards COVID-19 vaccine administration [21,22]. The main reasons for the reported hesitancy among the healthcare professionals comprised, rapid development of the vaccine, lack of safety and efficacy evidence, the vaccine does not prevent COVID-19 infection, and the fear of the unknown adverse effects [21]. In the UAE, healthcare workers were in the eye of the storm and witnessed the severity of the pandemic, which indeed positively affected their perception and acceptance of children vaccination against COVID-19, however the vaccine safety and adverse effects in children (>70%) are still of concern. Moreover, in the UAE, healthcare workers vaccination is mandatory, which provided an evidence on the safety of the vaccine in addition to the milder symptoms experienced with vaccinated individuals upon infection compared to unvaccinated people.

The UAE culture respects old people and encourages the whole society to be mindful of the elderly population and help and support them. Participants in this study who have an elderly family member within their household were found to be more prone to vaccinate their children to protect vulnerable individuals, i.e., the children and the old people, from the infection. These results came in agreement with findings from other studies that reported that vaccination to protect beloved ones is among the top factors associated with vaccination acceptance [21].

The role of social media in disseminating positive information about vaccination and its effect in halting the virus progression cannot be ignored. Its effective role in increasing public awareness about childhood vaccination has participated in the increase in parents’ confidence in vaccine safety and efficacy, which was reported in this study and previous ones [23]. However, the fact that social media can also be a source of misleading information should not be neglected, which may lead to an increase in public fears and misconception, posing more burdens and challenges for COVID-19 vaccination programs and recalling the critical role of governments in reinforcing the law governing the news published on the social media platforms.

Most parents who were willing to vaccinate their children below five years old were in Abu Dhabi, the capital of the UAE. Abu Dhabi was reported to be the highest emirate to accept childhood vaccination, with an approximate 76% acceptance rate. The remarkably high rate of acceptance is the result of extensive efforts in raising the awareness among the population and the strict regulations enforced by Abu Dhabi government in collaboration with the healthcare authorities. UAE’s outstanding efforts in implementing fast and effective COVID-19 preventive and curative measures have resulted in low infection and mortality rates compared to other countries worldwide [5].

The persistent efforts of the UAE government in educating and increasing the public awareness of the latest preventive measures against COVID-19 infection [21] have positively affected the parents’ beliefs in wearing face masks and children vaccination, as per the current study. These observations support data from previous studies that reported that public beliefs and awareness about COVID-19 vaccinations are greatly associated with vaccine acceptance in different countries such as USA [24], UK [25], France [26], and Italy [27]. This conclusion can help developing appropriate strategies and plans for implementing children vaccination programs, particularly in societies with low vaccination rate.

The literature has reported that proper interventions could assist in improving COVID-19 vaccine perceptions. Therefore, targeting and initiating interventions towards correcting all false perceptions about COVID-19 vaccines should be prioritized among governments and health authorities to effectively improve vaccination acceptance and rates [28].

Reluctance to accept vaccination against COVID-19 poses a significant health risk to the public, including children, and is known to be a multi-determined phenomenon. This hesitancy and refusal threaten the ability to establish herd immunity and negatively affect the global efforts to control and prevent COVID-19 infection. The current study reported that parents’ reluctance to vaccinate their children was low, due to different factors, however it is still a problem for many societies with low acceptance rate of vaccination. Furthermore, the findings imply that more work should be conducted to understand parents’ fears regarding the vaccine, including the factors that are involved in parents accepting the vaccine for their children.

The COVID-19 virus has not only disrupted the healthcare systems worldwide, but also greatly affected the educational systems and forced students to shift to online learning mode. The current study exploits the relationship between parents’ willingness to vaccinate their children against COVID-19 virus and their desire to send their children back to schools. The results showed that parents of children with poor academic achievement during the pandemic due to online learning are more inclined to vaccinate their children against the virus (*p* > 0.01).

Interestingly, the current study correlated the level of education of the participants with their acceptance of children vaccination, a direct relationship was observed as expected in a country such as the UAE, with a very high rate of literacy (96%). These observations are consistent with the findings of other studies in different countries such as the USA [17], Saudi Arabia [18], Ecuador, France, Germany, and India. The strong correlation between the educational level and vaccine acceptance can be explained by the higher public awareness in such well-educated countries [29].

Resuming the face-to-face educational modality and children resuming physical attendance in schools have driven the parents to accept the vaccination programs for children above 5 years old. Approximately 92% of the participants believe that their children’s academic performance was negatively impacted by online learning and hope that the vaccine could help in resuming physical attendance in schools. Schools’ management personnel have a significant role in raising the parents’ and children awareness about the importance of children vaccination against COVID-19.

## 5. Strengths and Limitations

To our knowledge, this is the first national study with a representative sample size in the UAE to investigate parents’ acceptance and willingness towards children vaccination and the associated factors.

The study has some limitations, the first limitation is the use of an online survey was primarily based on parents’ self-reports, including possible selection bias for participants in various socio-demographic characteristics. The second limitation is the unequal sample sizes taken from each emirate (city). Despite the limitations this national cross-sectional study is representative of the population from all emirates, with a sample size above the minimum required sample size. In addition, the results represented people with various socio-demographic characteristics such as age category, educational levels, and employment status (health and non-health sectors).

## 6. Conclusions

There is no doubt that the COVID-19 vaccines, like other vaccines, are effective interventions that can reduce or halt COVID-19 progression. However, public acceptance of children vaccination against COVID-19 has proven to be more complex in nature compared to traditional vaccines and is greatly affected by various factors and perceptions in each community. The rapid development of the vaccines and their long term effects on our children’s health were the major factors affecting the parents’ perception and belief in the available vaccination programs. Identifying the factors and the beliefs behind discouragement of child vaccination in the UAE is of a paramount importance. It will assist in developing and adopting proper effective interventions to improve public vaccination acceptance rates for children. Finally, the parents belief that their children’s academic performance was negatively affected by online learning has encouraged parents to accept the COVID-19 vaccination programs for children over the age of 5 years.

## Figures and Tables

**Table 1 vaccines-10-01434-t001:** Patients’ characteristics (*n* = 1049).

Socio-Demographic Characteristics	*n* (%)
Gender	
Male	293 (27.9)
Female	756 (72.1)
Age categories	
18–30	336 (32.0)
31–40	309 (29.5)
41–50	296 (28.2)
≥51	108 (10.3)
Residence area	
Abu Dhabi	420 (40.0)
Ajman	129 (12.3)
Al Ain	125 (11.9)
Dubai	177 (16.9)
Fujairah	26 (2.5)
Ras Al Khaimah	38 (3.6)
Sharjah	107 (10.2)
Um Quwain	27 (2.6)
Educational level	
High school	6 (0.6)
Undergraduate students	178 (17.0)
Bachelor	531 (50.6)
Postgraduate master	185 (17.6)
Postgraduate PhD	149 (14.2)
Employment status	
Healthcare worker	180 (17.2)
Non-healthcare worker	514 (49.0)
Retired	35 (3.3)
Student	198 (18.9)
Unemployed	122 (11.6)
Do your elderly live with you?	
No	496 (47.3)
Yes	553 (52.7)

**Table 2 vaccines-10-01434-t002:** Parents’ perspectives towards COVID-19 vaccine (*n* = 1049).

About the Vaccine	*n* (%)
Has any member of your family aged 16 taken the vaccine?	
No	274 (26.3)
Yes	769 (73.7)
Have your children or siblings aged below 16 taken the vaccine?	
No	601 (57.3)
Yes	447 (42.6)
If the COVID-19 vaccine is available and affordable for children above 5, will you go for it?	
No	307 (29.3)
Yes	742 (70.7)
If your answer is “No”, what best describes the reason why?	
I am concerned about side effects and safety	229 (74.6)
I am concerned about the speed it was made	43 (14)
I do not think they need it	35 (11.4)
Have you or someone you know been infected by COVID-19?	
No	365 (34.8)
Yes	684 (65.2)
Have any of your children contracted COVID-19?	
No	582 (55.5)
Yes	467 (44.5)
Does your child/children attend school face-to-face or online?	
Online	809 (77.1)
Direct	240 (22.9)
**Did distance learning affect their academic scores?**	
No	369 (35.2)
Yes	680 (64.8)
Do you plan to return your child/children back to face-to-face learning if this is possible?	
No	260 (25.1)
Yes	776 (74.9)
If your child/children attend school physically, to what extent do you feel they are in danger in a score from 0–5? As zero means no risk at all and five means the highest risk possible.	
0.0	25 (2.5)
1.0	108 (10.7)
2.0	195 (19.2)
3.0	262 (25.8)
4.0	137 (13.5)
5.0	287 (28.3)
Do you believe that vaccination of children will help schools to return to face-to-face education?	
No	305 (29.1)
Yes	744 (70.9)
Do you believe that vaccination of children will protect the children more?	
No	219 (20.9)
Not sure	282 (26.9)
Yes	548 (52.2)
Are you still generally concerned or worried about the COVID-19 pandemic?	
No	306 (29.2)
Yes	743 (70.8)
Did your child/children receive a flu shot?	
No	434 (41.4)
Yes	615 (58.6)
Does/do your child/children usually wear a mask or face covering when in public?	
No	176 (16.8)
Yes	873 (83.2)
Do you usually keep your child/children at home when they feel slightly unwell?	
No	223 (21.3)
Yes	826 (78.7)
What is your source of information about the COVID-19 vaccination scheme for children?	
Families	56 (5.3)
Friends	139 (13.3)
Healthcare bodies	172 (16.4)
Newsletter	3 (0.3)
Published articles	15 (1.4)
Social media	528 (50.3)
TV	66 (6.3)
WHO	70 (6.7)

**Table 3 vaccines-10-01434-t003:** Association between socio-demographic characteristics and parents’ acceptance of the COVID-19 vaccine (*n* = 1049).

Socio-Demographic Characteristics	If the COVID-19 Vaccine Is Available and Affordable for Children above 5, Will You Go for It?		*p* Value
	No *N* (%)	Yes *N* (%)	
Gender			<0.001
Male	46 (15.7)	247 (84.3)	
Female	261 (34.5)	495 (65.3)	
Age categories			<0.001
18–30	72 (21.4)	264 (78.6)	
31–40	86 (27.8)	223 (72.2)	
41–50	108 (36.5)	188 (63.5)	
≥51	41 (38.0)	67 (62.0)	
Residence area			<0.001
Abu Dhabi	102 (24.3)	318 (75.7)	
Ajman	47 (36.4)	82 (63.6)	
Al Ain	54 (43.2)	71 (56.8)	
Dubai	48 (27.1)	129 (72.9)	
Fujairah	3 (12.0)	22 (88.0)	
Ras Al Khaimah	18 (47.4)	20 (52.6)	
Sharjah	30 (28.0)	77 (72.0)	
Um Quwain	5 (18.5)	22 (81.5)	
Educational level			0.003
Bachelor	179 (33.7)	352 (66.3)	
High school	0 (0.0)	6 (100.0)	
Undergraduate students	37 (20.8)	141 (79.2)	
Postgraduate master	55 (29.7)	130 (70.3)	
Postgraduate PhD	36 (24.2)	113 (75.8)	
Employment status			<0.001
Healthcare worker	24 (13.3)	156 (86.7)	
Non-healthcare worker	179 (34.8)	335 (65.2)	
Retired	12 (34.3)	23 (65.7)	
Student	47 (23.7)	151 (76.3)	
Unemployed	45 (36.9)	77 (63.1)	

Chi square.

**Table 4 vaccines-10-01434-t004:** Association between parents’ perception towards COVID-19 vaccine and parents’ acceptance of the COVID-19 vaccine (*n* = 1049).

About the Vaccine	If the COVID-19 Vaccine is Available and Affordable for Children above 5, Will You Go for It?		*p* Value
	No *N* (%)	Yes *N* (%)	
Has any member of your family aged above 16 taken the vaccine?			<0.001
No	178 (65.0)	96 (35.0)	
Yes	129 (16.8)	640 (83.2)	
Have your children aged below 16 taken the vaccine?			<0.001
No	261 (43.4)	340 (56.6)	
Yes	46 (10.3)	401 (89.7)	
Do your elderly live with you?			<0.001
No	184 (37.1)	312 (62.9)	
Yes	123 (22.2)	430 (77.8)	
Have you or someone you know contracted COVID-19?			<0.001
No	189 (51.8)	176 (48.2)	
Yes	118 (17.3)	566 (82.7)	
Have any of your children contracted COVID-19?			<0.001
No	232 (40.5)	341 (59.5)	
Yes	73 (15.6)	394 (84.4)	
Do you believe that vaccination of children will help schools to return to face-to-face education?			<0.001
No	235 (77.0)	70 (23.0)	
Yes	72 (9.7)	672 (90.3)	
Are you still generally concerned or worried about the COVID-19 pandemic?			<0.001
No	177 (57.8)	129 (42.2)	
Yes	130 (17.5)	613 (82.5)	
Do you believe that vaccination of children will protect the children more?			<0.001
No	200 (91.3)	19 (8.7)	
Yes	107 (12.9)	723 (87.1)	
Did your child/children receive a flu shot?			<0.001
No	207 (47.7)	227 (52.3)	
Yes	100 (16.3)	515 (83.7)	
Does/do your child/children usually wear a mask or face covering when in public?			<0.001
No	136 (81.0)	32 (19.0)	
Yes	169 (19.4)	704 (80.6)	
Do you usually keep your child/children at home when they feel slightly unwell?			<0.001
No	160 (71.7)	63 (28.30	
Yes	147 (17.8)	679 (82.2)	
What is your source of information about the COVID-19 vaccination scheme for children?			<0.001
Families	33 (60.0)	22 (40.0)	
Friends	32 (23.0)	107 (77.0)	
Healthcare bodies	44 (26.3)	123 (73.7)	
Newsletter	3 (100.0)	0 (0.0)	
Published articles	9 (60.0)	6 (40.0)	
Social media	145 (27.5)	383 (72.5)	
TV	20 (30.3)	46 (69.7)	
WHO	20 (28.6)	50 (71.4)	

**Table 5 vaccines-10-01434-t005:** Association between socio-demographic characteristics and parents’ willingness to get their children back to school (*n* = 1049).

Socio-Demographic Characteristics of Parents Whose Children Had Online Learning	Do you Plan to Return Your Child/Children Back to Face-to-Face Learning if this Is Possible?		*p* Value
	No *N* (%)	Yes *N* (%)	
Gender			0.026
Male	51 (24.1)	157 (74.1)	
Female	203 (34.0)	385 (64.5)	
Age			<0.001
18–30	67 (25.6)	188 (71.8)	
31–40	51 (22.1)	177 (76.6)	
41–50	89 (38.5)	141 (61.0)	
>51	47 (55.3)	36 (42.4)	
Residence area			<0.001
Abu Dhabi	60 (18.1)	263 (79.5)	
Ajman	41 (41.4)	58 (58.6)	
Al Ain	55 (53.9)	43 (42.2)	
Dubai	42 (35.3)	76 (63.90	
Fujairah	14 (63.6)	8 (36.4)	
Ras Al Khaimah	12 (41.4)	17 (58.6)	
Sharjah	23 (28.4)	58 (71.6)	
Um Quwain	7 (26.9)	19 (73.1)	
Educational level			<0.001
Bachelor	121 (29.1)	284 (68.3)	
High school	6 (100.0)	0 (0.0)	
Undergraduate	43 (30.9)	95 (68.3)	
Postgraduate master	55 (39.0)	86 (61.0)	
Postgraduate PhD	29 (27.1)	77 (72.0)	
Study field			<0.001
Medical	27 (10.5)	226 (88.3)	
Non-medical	227 (41.0)	316 (57.1)	
Employment status			<0.001
Healthcare worker	15 (12.8)	99 (84.6)	
Non-healthcare worker	160 (38.6)	249 (60.1)	
Retired	9 (32.1)	19 (67.9)	
Student	49 (32.0)	103 (67.3)	
Unemployed	21 (21.6)	72 (74.2)	

Chi square.

**Table 6 vaccines-10-01434-t006:** Association between parents’ perspectives towards COVID-19 vaccine and parents’ willingness to get their children back to school (*n* = 1049).

Vaccine Perception for Patients Whose Children Had Online Learning	Do You Plan to Return Your Child/Children Back to Face-to-Face Learning if this Is Possible?		*p* Value
	No *N* (%)	Yes *N* (%)	
If the COVID-19 vaccine is available and affordable for children above 5, will you go for it?			<0.001
No	173 (65.5)	87 (33.0)	
Yes	81 (14.9)	455 (83.5)	
If online, did distance learning affect their academic scores?			<0.001
No	223 (69.0)	97 (30.0)	
Yes	31 (6.5)	442 (92.1)	
If your child/children attend school physically, to what extent do you feel they are in danger in a score from 0–5? As zero means no risk at all and five means the highest risk possible.			<0.001
0	0 (0.0)	13 (100.)	
1	3 (5.3)	54 (94.7)	
2	2 (1.6)	121 (93.8)	
3	12 (6.1)	184 (93.4)	
4	8 (7.5)	99 (92.5)	
5	225 (82.1)	48 (17.5)	
Do you believe that vaccination of children will help schools to return to face-to-face education?			<0.001
No	217 (78.1)	59 (21.2)	
Yes	37 (7.0)	483 (91.0)	
Do you believe that vaccination of children will protect the children more?			<0.001
No	166 (85.6)	28 (14.4)	
Not sure	49 (20.4)	184 (76.7)	
Yes	39 (10.4)	330 (88.0)	
Do you usually keep your child/children at home when they feel slightly unwell?			<0.001
No	160 (74.8)	54 (25.2)	
Yes	94 (15.8)	488 (82.0)	
Does/do your child/children usually wear a mask or face covering when in public?			<0.001
No	135 (81.3)	31 (18.7)	
Yes	119 (18.7)	511 (80.1)	
Did your child/children receive a flu shot?			<0.001
No	180 (49.5)	180 (49.5)	
Yes	74 (16.6)	362 (81.3)	
Are you still generally concerned or worried about the COVID-19 pandemic?			<0.001
No	164 (63.6)	92 (35.7)	
Yes	90 (16.3)	450 (81.7)	
Have any of your children contracted COVID-19?			<0.001
No	204 (41.0)	289 (58.0)	
Yes	50 (16.6)	248 (82.1)	
Have you or someone you know contracted COVID-19?			<0.001
No	163 (52.8)	144 (46.6)	
Yes	91 (18.2)	398 (79.6)	

**Table 7 vaccines-10-01434-t007:** Logistic regression taking the parents’ acceptance of the COVID-19 vaccine as the dependent variable and all other factors as the independent variables (*n* = 1049).

Variables	OR	95% Confidence Interval	*p* Value
Gender (Male vs. Female)	0.403	0.229–0.709	0.002
Residence area			
Al Ain vs. Abu Dhabi	0.633	0.339–1.180	0.150
Dubai vs. Abu Dhabi	2.054	1.016–4.151	0.45
Fujairah vs. Abu Dhabi	9.129	1.526–54.635	0.015
Ras Al Khaimah vs. Abu Dhabi	0.309	0.90–1.065	0.063
Sharjah vs. Abu Dhabi	1.094	0.530–2.261	0.808
Um Quwain vs. Abu Dhabi	2.833	0.418–19.193	0.286
Ajman vs. Abu Dhabi	0.659	0.331–1.312	0.253
Employment status			
Non-healthcare worker vs. healthcare worker	0.539	0.255–1.139	0.106
Retired vs. healthcare worker	0.231	0.058–0.929	0.039
Student vs. healthcare worker	0.339	0.151–0.764	0.009
Unemployed vs. healthcare worker	0.269	0.115–0.626	0.002
Has any member of your family aged above 16 taken the vaccine?			
(Yes vs. no)	1.718	1.057–2.792	0.029
Have your children or siblings aged below 16 taken the vaccine?			
(Yes vs. no)	2.205	1.333–3.648	0.002
Do you believe that vaccination of children will protect the children more?			
(Yes vs. no)	37.742	19.961–71.360	<0.001
Did your child/children receive a flu shot?			
(Yes vs. no)	1.586	1.012–2.487	0.044
Does/do your child/children usually wear a mask or face covering when in public?			
(Yes vs. no)	3.038	1.587–5.815	0.001

**Table 8 vaccines-10-01434-t008:** Logistic regression taking the parents’ willingness to get their children back to school as the dependent variable and all other factors as the independent variables (*n* = 1049).

Variables	OR	95% Confidence Interval	*p* Value
Age categories			
31–40 vs. 18–30	0.357	0.077–1.646	0.186
41–50 vs. 18–30	0.176	0.039–0.792	0.024
>51 vs. 18–30	0.065	0.011–0.380	0.002
Educational level			
High school vs. undergraduate	0.000	0.000–0.0	0.999
Bachelor vs. undergraduate	4.704	1.211–18.271	0.025
Postgraduate master vs. undergraduate	2.182	0.399–11.925	0.368
Postgraduate PhD vs. undergraduate	14.855	2.525–87.386	0.003
Employment status			
Non-healthcare worker vs. healthcare worker	0.260	0.055–1.225	0.088
Retired vs. healthcare worker	0.465	0.047–4.610	0.513
Student vs. healthcare worker	0.169	0.019–1.525	0.113
Unemployed vs. healthcare worker	0.996	0.167–5.944	0.997
Did distance learning affect their academic scores?			
(Yes vs. no)	3.040	1.373–6.727	0.006
Do you believe that vaccination of children will help schools to return to face-to-face education?			
(Yes vs. no)	14.993	6.677–33.669	<0.001
**Did your child/children receive a flu shot?**			
(Yes vs. no)	4.509	1.937–10.497	<0.001
Does/do your child/children usually wear a mask or face covering when in public?			
(Yes vs. no)	4.577	1.297–16.150	0.018
Do you usually keep your child/children at home when they feel slightly unwell?			
(Yes vs. no)	2.587	0.891–7.512	0.081

## Data Availability

Data will be available upon request.

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
