# Peer review of "Parents’ Perception, Acceptance, and Hesitancy to Vaccinate Their Children against COVID-19: Results from a National Study in the UAE"

_vaccines, 2022, doi:10.3390/vaccines10091434_

Round 1

Reviewer 1 Report

Thank-you for this paper that addresses an important area.

I have a few suggestions:

1.  In the introduction, can you explain why it was necessary to carry out the study in UAE?   why would your findings be expected to be different to 2.  studies already carried out in similar countries?

2.  Your results are broader than your study aims.  Consider having one overall study aim that incorporates all aspects of your study.

3.  In the data analysis subsection of your methods, please include your methodology for the multi-variate tests.

4.  In the validity subsection of the methods, please include any methods you used to ensure validity of translated version e.g. back translation.

5.  In the results section,  please avoid repetition between text and tables.  Instead, use the text to refer to the tables and to draw out the key findings of the results.   

6.  Consider structuring your discussion section more e.g. to include a brief summary of your findings, comparison to other literature, implications for practice and policy, and then the strengths and limitations section.  Try and be clear throughout on what your study has specifically added to previous literature.

Author Response

Dear Reviewer

Thank you very much for valuable comments and suggestions which added a lot to the study. Please find below answers for the comments and suggestions. Also, kindly find attached the revised manuscript.  

Best regards

Comment 1: In the introduction, can you explain why it was necessary to carry out the study in UAE? why would your findings be expected to be different to 2 studies already carried out in similar countries?

Answer: The importance of carrying out the study in UAE was mentioned, please see the manuscript [Lines 84-97 and 115-119], with some addition for this necessary of the study: the study will provide important data to the scientific community and the healthcare sector in the UAE and will contribute to identifying the prevalence of the vaccinated children and its impact on the choice of the educational modality in future. This study has a unique aim, which do not study in previous studies: “the link with the choice of distance learning instead of face-to-face education by the parents that covering all the outcomes”.

Comment 2:  Your results are broader than your study aims. Consider having one overall study aim that incorporates all aspects of your study.

Answer: Thanks for this good comment. The study has two main interrelated aims: [1] parent’ acceptance, perception and hesitancy toward the COVID-19 vaccine administration for their children and [2] the link with the choice of distance learning instead of face-to-face education by the parents that covering all the outcomes. In addition, there are many variables assessed in the study such as age, medical background, education status, living area, etc. Some results are directly related to one goal, and others may support them. Therefore, the outcomes might be big, however, the outcomes are related to the two aims.  

Comment 3:  In the data analysis subsection of your methods, please include your methodology for the multi-variate tests.

Answer: paragraph about Multivariate analysis was added to the method section (Statistical Analysis, page 5, lines 209-214), see in the revised manuscript.

Comment 4:  In the validity subsection of the methods, please include any methods you used to ensure validity of translated version e.g. back translation.

Answer: Method of developing and validation of Arabic questionnaire, including translation, was added to page 4 (lines 192-196), see in the revised manuscript.

Comment 5:  In the results section, please avoid repetition between text and tables. Instead, use the text to refer to the tables and to draw out the key findings of the results.

Answer: Some paragraphs related to results section are deleted to avoid repetition between text and tables, see results section at the revised manuscript.

Comment 6:  Consider structuring your discussion section more e.g. to include a brief summary of your findings, comparison to other literature, implications for practice and policy, and then the strengths and limitations section.  Try and be clear throughout on what your study has specifically added to previous literature.

Answer: the discussion section was modified and a brief summary of findings was added, see discussion section at the revised manuscript.

Reviewer 2 Report

This is a very interesting research paper that follows the title of the study.

Some of the problems and modifications are listed below.

Abstract

No changes are necessary.

Introduction

No particular modifications are needed.

Materials and Methods

No special modifications are needed.

Results

3.1. Socio-demographic Characteristics

Is "with 72%) males" a typo? The results in the table show that 72.1% of the respondents are female.

In Table 1, does Undergraduate mean currently enrolled?

If so, are student marriages and student births more common in the UAE?

Discussion

The first paragraph is almost exactly what should be discussed in the Introduction

In the UAE Acceptance of vaccination of children is higher among health care workers

COVID-19 practices on the front lines is what we are doing all over the world.

Please explain why this is so.

Conclusions

Vaccination as a method of acquire immunity has proven it efficacy since hundreds of years in controlling virulent viruses, preventing, and eliminating of pandemics.”  is not a result of this study.

This content should be excluded from the Conclusions.

"The aims of this study can be summarized as follows [1] explore parents' acceptance, perception, and hesitancy toward the COVID-19 vaccine administration for their children, and [2] link parents' vaccine acceptance to their choice of distance versus Conclusions

Please provide a brief description of the results for this objective.

Author Response

Dear Reviewer

Thank you very much for valuable comments and suggestions which added a lot to the study. Please find below answers for the comments and suggestions. Also, kindly find attached the revised manuscript.  

Best regards

Comment 1: 3.1. Socio-demographic Characteristics

Is "with 72%) males" a typo? the results in the table show that 72.1% of the respondents are female.

Answer: This is a mistake. The results have been corrected to be female.

Comment 2: In Table 1, does Undergraduate mean currently enrolled? If so, are student marriages and student births more common in the UAE?

Answer: Result corrected to be undergraduate students, see in the revised manuscript (Tables 1, 3 and 5). In the UAE, some people (specifically females) are married at the age of 18 years, before attending college.

Discussion

Comment 3: The first paragraph is almost exactly what should be discussed in the Introduction.

Answer: Part of the paragraph was deleted and another part was moved to be in the introduction section, see in the revised manuscript (introduction section, lines 69 – 82).

Comment 4: In the UAE, acceptance of vaccination of children is higher among health care workers COVID-19 practices on the front lines is what we are doing all over the world. Please explain why this is so.

Answer: Done, paragraph was added (page 16, 370-374)

Conclusions

Comment 5: “Vaccination as a method of acquire immunity has proven it efficacy since hundreds of years in controlling virulent viruses, preventing, and eliminating of pandemics”, is not a result of this study. This content should be excluded from the Conclusions.

Answer: The paragraph deleted from the conclusion section, see the revised manuscript.

Comment 6: "The aims of this study can be summarized as follows [1] explore parents' acceptance, perception, and hesitancy toward the COVID-19 vaccine administration for their children, and [2] link parents' vaccine acceptance to their choice of distance versus Conclusions. Please provide a brief description of the results for this objective.

Answer:

For aim [1], it is written in the conclusion.

For aim [2], a paragraph was added (page 18, lines 365 - 467). 

Reviewer 3 Report

I suggest that tables 1 to 6 include the 95% confidence intervals. It is suggested to report all percentages or values to one or a maximum of two decimal places, only the value of P should have three decimal places. Results and conclusions. Considering the methodological implications of a national study, it is not possible to interpret the results and discuss them in that context. The limitations and strengths of the study should be delved into, as well as the possible inferences of the results at the national level. Since it is possible to identify a possible selection bias. It is unknown if the distribution of the study sample and the population distribution in the areas where it was carried out, as well as its socio-demographic characteristics, age, etc., are at least proportional.

Author Response

Dear Reviewer

Thank you very much for valuable comments and suggestions which added a lot to the study. Please find below answers for the comments and suggestions. Also, kindly find attached the revised manuscript.  

Best regards

Comment 1: I suggest that tables 1 to 6 include the 95% confidence intervals.

Answer: For Tables 1 to 6, the written results did not include “Means” because the answers were Yes or No; therefore, 95% CI cannot be used in these tables.

Comment 2: It is suggested to report all percentages or values to one or a maximum of two decimal places, only the value of P should have three decimal places.

Answer: Most of percentages value were modified to one decimal only (in the manuscript and tables), except for Tables 6 and 7 and paragraphs related to them with three decimals (for statistics reason), see in the revised manuscript.

Comment 3: Results and conclusions. Considering the methodological implications of a national study, it is not possible to interpret the results and discuss them in that context. The limitations and strengths of the study should be delved into, as well as the possible inferences of the results at the national level. Since it is possible to identify a possible selection bias. It is unknown if the distribution of the study sample and the population distribution in the areas where it was carried out, as well as its sociodemographic characteristics, age, etc., are at least proportional.

Answer: Some modifications have been made to the results and conclusions sections to make them possible to interpret the results and then discuss them in the desired context, see in the revised manuscript (results and conclusion sections). The strength and limitation part was re-written to include the mentioned points.

Round 2

Reviewer 1 Report

Thank-you for making some revisions.  I have the following suggestions:

1.   You have explained why the results would be important for a UAE audience but 'Vaccines' is an international journal.   Please emphasise more in the introduction and discussion the novelty of your study and findings for an international audience.  You may want to draw more on the educational delivery associations.

2.  Please state one overall aim for your study.

3.  Please state in your methods section that you used logistic regression.

3.  Please further reduce the text in the results as it is still very repetitive of the tables.

4.  Please include an implications for practice section in the discussion.

Author Response

Dear Reviewer 

Thanks for valuable comments and suggestions. Please see below our answers. 

Comment 1: You have explained why the results would be important for a UAE audience but 'Vaccines' is an international journal. Please emphasize more in the introduction and discussion the novelty of your study and findings for an international audience. You may want to draw more on the educational delivery associations.

Answer: Done, please see introduction section (at end), page 3, and in discussion section, page 17.

Comment 2:  Please state one overall aim for your study.

Answer: Done, please see introduction section (at end), page 3.

Comment 3:  Please state in your methods section that you used logistic regression.

Answer: the statement was added (Method, Statistical Analysis, page 5)

Comment 4:  Please further reduce the text in the results as it is still very repetitive of the tables.

Answer: Further reducing the text in the results was done. Please see results section at the manuscript.

Comment 5:  Please include an implication for practice section in the discussion.

Answer: please see discussion section, mainly page 17.

Best regards

Reviewer 3 Report

Thanks for your reply 

For table 1 to 6, it will be highly recommended to report  95%CI for percentages

https://ebn.bmj.com/content/15/3/66

Author Response

Dear Reviewer 

Thank you for your valuable comments.

Table 1 and 2 represent descriptive results for categorical variables, so frequencies and percentages describe these variables.

For tables 3 to 6 present bivariate results (chi square test) in which frequencies and percentages and P values represent the association between COVID-19 vaccination and other variables.

However table 7 and 8 represent logistic regression results and it include confidence interval with P value as this table assess the association between independent and dependent variables.

Best regards